analytical chemistry/environmental chemistry/environmental science

cadmium, X-ray fluorescence, algae, preconcentration, rapid detection

**Author for correspondence:**
Nanjing Zhao
e-mail: njzhao@aiofm.ac.cn

# Preconcentration with *Chlorella vulgaris* combined with energy dispersive X-ray fluorescence spectrometry for rapid determination of Cd in water

Tingting Gan[1,2], Nanjing Zhao[1,2], Gaofang Yin[1,2], Min Chen[1,2,3], Xiang Wang[1,2,3] and Hui Hua[1,2,3]

[1]Key Laboratory of Environmental Optics and Technology, Anhui Institute of Optics and Fine Mechanics, Chinese Academy of Sciences, Hefei 230031, People's Republic of China
[2]Key Laboratory of Optical Monitoring Technology for Environment, Anhui Province, Hefei 230031, People's Republic of China
[3]University of Science and Technology of China, Hefei 230026, People's Republic of China

TG, 0000-0001-7170-9680

Freshwater green algae *Chlorella vulgaris* was selected as an adsorbent, and a simple, rapid, economical and environmentally friendly method for the detection of heavy metal Cd in water samples based on preconcentration with *C. vulgaris* combined with energy dispersive X-ray fluorescence (EDXRF) spectrometry was proposed. *Chlorella vulgaris* could directly and rapidly adsorb $Cd^{2+}$ without any pretreatment, and the maximum adsorption efficiency could be obtained when the contact time was 1 min with an optimal pH of 10. The obtained Cd-enriched thin samples after preconcentration with *C. vulgaris* by suction filtration of reaction solution had very good uniformity, which could be directly measured by EDXRF spectrometry, and the net integral fluorescence intensity of Cd K$\alpha$ characteristic peak had a very good linear relationship with the initial concentration of Cd in the range of 0.703–74.957 µg ml$^{-1}$ with a correlation coefficient of 0.9979. When the Cd thin samples with a Cd-enriched region of 15.1 mm in diameter were formed by the developed preconcentration method with suction filtration of 10 ml reaction solution, the detection limit of this method was 0.0654 µg ml$^{-1}$, which was lower than the maximum allowable discharge concentration of Cd in various industrial wastewaters. The proposed method was simple to operate, and could effectively remove the influence of matrix effect of water

This article has been edited by the Royal Society of Chemistry, including the commissioning, peer review process and editorial aspects up to the point of acceptance.

samples and effectively improve the sensitivity and stability of EDXRF spectrometry directly detecting heavy metals in water samples, which was successfully applied to detect Cd in real water samples with satisfactory results, and the recoveries ranged from 94.80% to 116.94%. Moreover, this method can be applied to the rapid detection and early warning of excessive Cd in discharged industrial wastewaters. This work will provide a methodological basis for the development of rapid and online monitoring technology and instrument of heavy metal pollutants in water.

## 1. Introduction

Wastewaters discharged from various industrial processes such as plating, mining, smelting, dyeing, battery, chemical industry and sewage discharged from agricultural activities usually contain large amounts of heavy metals, including cadmium (Cd). Cd is one of the most toxic heavy metal environmental pollutants. Due to its characteristics of non-biodegradation, bioaccumulation and concealing, Cd is potentially dangerous for human beings, animals and plants. Once it enters the human body through the food chain, the haematopoiesis, nerves, liver, kidneys, respiratory system, cardiovascular system and reproductive system of human body will be seriously affected [1], which will cause acute and chronic diseases and seriously endanger human health and safety. In addition, Cd has been known as a human carcinogen by the International Agency for Research on Cancer [2,3]. Therefore, the rapid, real-time and online detection of Cd in water has very important practical significance for preventing water pollution from heavy metals and protecting the safety of human beings and ecological environment.

The conventional detection methods of Cd, such as atomic absorption spectrometry [4,5], inductively coupled plasma atomic emission spectrometry [6,7] and inductively coupled plasma mass spectrometry [8,9], although having their own advantages, such as high sensitivity and good accuracy, most of them require bulky and expensive equipment, complicated pretreatment procedures and professional technicians to operate them. The entire detecting processes are usually sophisticated and time consuming. Therefore, these methods cannot meet the requirements for rapid, real-time, online detection of Cd in water.

Compared with the above-mentioned conventional methods, X-ray fluorescence spectroscopy (XRF) is an important spectroscopy technology for the rapid detection of heavy metals. Because of the advantages of non-destructiveness for the sample, simultaneous determination of multiple elements, less spectral line interference, fast measurement speed and good reproducibility [10], XRF spectrometry has been an effective method for rapid, real-time and online analysis of heavy metal elements. However, when XRF spectrometry is directly applied to detect heavy metals in water samples, there are two main problems: The first one is when aqueous solution is irradiated by high energy X-rays, some chemical reactions will happen between elements in the irradiated part due to local heating, at the same time some phenomena such as bubbling and precipitation will appear, which makes the X-ray fluorescence intensity very easy to change and seriously affects the reproducibility and stability of the analytical signal. Secondly, due to the strong absorption capacity of aqueous solution for X-rays, the spectral line intensity of analysed elements will be weakened and at the same time there will be a strong X-ray scattering background [11]. Consequently, for the direct detection of heavy metals in water samples, XRF spectrometry has the disadvantages of low signal-to-noise ratio [12], matrix effects [13], poor sensitivity and poor stability, which cannot meet the requirements for rapid detection of heavy metals in environmental waters and industrial wastewaters.

In recent years, some preconcentration methods of heavy metals in water such as solid phase extraction have been used prior to XRF spectrometry measurement in order to improve the shortcomings of XRF spectrometry directly used for the detection of heavy metals in water samples. For the solid phase extraction preconcentration method, a solid adsorbent will be used to adsorb and extract the analysed heavy metals and transfer them from liquid phase to solid phase. Because the method can effectively remove the complex matrix of water samples and separate the target heavy metals, so the scattering background of the XRF spectrum is greatly reduced, and the signal-to-noise ratio is effectively improved. In this way, the precision and sensitivity of XRF spectrometry are effectively improved. So far, the solid adsorbents used in the solid phase extraction preconcentration method mainly include activated carbon [14], carbon nanotube [15], graphene [16–19] and resin [20–22]. For activated carbon, carbon nanotube and graphene, in order to improve the adsorption capability for heavy metals, they usually need to be functionally modified by combining with some organic macromolecules before use. However, the process of functional modification is complicated, time consuming, and poorly repeatable. In addition, the use of organic macromolecules is likely to

cause secondary environmental pollution. For the resin, before being used for preconcentration of heavy metals in water samples, they need to go through a long pretreatment process including drying, comminution, grinding, sieving, expansion, acid soak and washing. The pretreatment process is also cumbersome and time consuming. Moreover, the use of expensive adsorbents such as activated carbon, carbon nanotube, graphene and resin will increase the application cost of XRF spectrometry.

At present, biosorption is a relatively novel method for treatment and removal of heavy metals from wastewater. Algae, as a kind of biosorbent, have the advantages of various species, easy to culture, easy to obtain, low cost and environmental friendliness. Moreover, for the adsorption of heavy metals, they also have the advantages of fast adsorption speed, large adsorption capacity and high adsorption efficiency. Therefore, in recent years, algae have attracted wide attention in treatment and repair of heavy metal-contaminated water. Many researchers have carried out some related research works in this area. For example, Khan *et al*. used four freshwater algae (*Cladophora glomerata*, *Oedogonium westii*, *Vaucheria debaryana* and *Zygnema insigne*) for remediating industrial wastewater containing potentially toxic elements such as Cd, Cr, Pb and Ni, and they found that *C. glomerata* had the highest removal capacity of 80.3% for Cd [23]. Henriques *et al*. used *Ulva lactuca* (green) to assess and explore the bioaccumulation and biosorption capability for Hg in contaminated waters with high salinity, and the effective removal could be up to 99% [24]. Consequently, algae are a kind of ideal adsorption biomaterial for heavy metals in water samples, which provides a new preconcentration idea for real-time and online monitoring of heavy metals in water samples by XRF spectrometry.

*Chlorella vulgaris* is a kind of common freshwater green algae. It has the characteristics of spherical or ellipsoidal cell shape, small cell size and large specific surface area, which is very beneficial for the adsorption of heavy metals from water samples. So in this study, *C. vulgaris* was selected as an adsorbent, and a simple and rapid method for the detection of Cd in water based on preconcentration with *C. vulgaris* combined with energy dispersive X-ray fluorescence (EDXRF) spectrometry was proposed. The effects of experimental conditions such as contact time, pH of reaction solution and initial $Cd^{2+}$ concentration on the adsorption efficiency of Cd by *C. vulgaris* were firstly investigated. Then the influences of different coexisting ions on the adsorption efficiency of Cd by *C. vulgaris* were investigated. Under the optimal adsorption conditions, the homogeneous Cd-enriched thin samples were obtained by suction filtration of reaction solution in order to collect the algae cells with adsorbed Cd on the mixed cellulose lipid microfiltration membranes. The obtained Cd thin samples could be directly measured by EDXRF spectroscopy, which effectively removed the influence of matrix effect of water samples and effectively improved the sensitivity and stability of EDXRF spectrometry directly detecting heavy metals in water samples. The proposed method for the determination of Cd in water samples is simple, rapid, economical and environmentally friendly, and had been successfully applied to detect Cd in real water samples, which can provide a methodological basis for the development of real-time, rapid and online monitoring technology and instrument of heavy metal pollutants in water.

# 2. Experimental

## 2.1. Algae culture and characterization

The freshwater green algae *C. vulgaris* used in this experiment were obtained from the Freshwater Algae Culture Collection at the Institute of Hydrobiology. BG11 medium containing $NaNO_3$, $K_2HPO_4$, $MgSO_4$, $CaCl_2 \cdot H_2O$, $Na_2CO_3$, citric acid, ammonium ferric citrate, EDTA and trace metals (e.g. $H_3BO_3$, $MnCl_2$, $ZnSO_4$, $Na_2MoO_4$, $Co(NO_3)_2$, $CuSO_4$,) for the inoculation and culture of *C. vulgaris* was prepared with ultrapure water. Prior to use, the medium and glass flasks used during inoculation and cultivation were sterilized by autoclaving in an autoclave at 121°C for 30 min. In a SW-CJ-1D ultra-clean work-bench (Shangyu Aike Instrument Equipment Co., Ltd, China), *C. vulgaris* were aseptically inoculated in 1000 ml glass flasks containing 400 ml BG11 medium and then cultured in a HP400G intelligent light incubator (Wuhan Ruihua Instrument and Equipment Co., Ltd, China). The culture conditions for *C. vulgaris* in the incubator were set as follows: the culture temperature was 25°, the light illumination was 60 µmol m$^{-2}$ s$^{-1}$, and the light : dark cycle was 12 h : 12 h. All flasks were shaken three times and the state of *C. vulgaris* culture solution was observed daily. When *C. vulgaris* were cultured for 8 days, *C. vulgaris* cells in the culture solution were characterized by a CX41 microscope (Olympus Corporation, Japan) and the pH value of *C. vulgaris* culture solution was measured by a PHS-3C pH meter (Leici Analytical Instrument Factory, China). The biomass of *C. vulgaris* in culture solution was determined as follows: First, a mixed cellulose lipid microfiltration membrane with a pore size of 0.22 µm and a diameter of 50 mm (Shanghai

Xingya Purification Material Factory, China) was weighed by a AL104 electronic scales (Mettler Toledo, China), then it and a SHB-IIIG vacuum pump (Zhengzhou Great Wall Technology Industry and Trade Co., Ltd, China) were used to filter 15 ml of *C. vulgaris* culture solution. After being air-dried, the microfiltration membrane enriched with algae cells was weighed again, and then according to the mass difference of the microfiltration membrane before and after enriching the algae cells and the volume of the filtered *C. vulgaris* culture solution, the biomass of *C. vulgaris* in culture solution was calculated. Then *C. vulgaris* in the culture solution were used for adsorption and preconcentration of Cd in water samples.

## 2.2. Preconcentration procedure

Ultrapure water with a resistivity of 18.25 MΩ cm obtained from a Millipore Milli-Q purification system (Molecular Corporation, Shanghai, China) was used during all the experiments. Cadmium Chloride (CdCl$_2$·2.5H$_2$O) used in the adsorption experiment was of analytical grade and purchased from Tianjin Guangfu Fine Chemical Research Institute (Tianjin, China). Cd$^{2+}$ stock solution with a concentration of 500 mg l$^{-1}$ was prepared using ultrapure water. Nitric acid (HNO$_3$) (67–70%) was of trace metal grade and purchased from Thermo Fisher Scientific (China) Co., Ltd. Sodium borate (Na$_2$B$_4$O$_7$·10H$_2$O) and sodium hydroxide (NaOH) were of analytical grade and purchased from Tianjin Recovery Fine Chemical Industry Research Institute (Tianjin, China). NaOH solution (0.1 mol l$^{-1}$), 2% (v/v) dilute HNO$_3$ solution and 0.05 mol l$^{-1}$ Na$_2$B$_4$O$_7$-NaOH buffer solution with a pH of 10 were respectively prepared to adjust the pH values of reaction solutions. All glass vessels were soaked in dilute HNO$_3$ solution (v/v, 5%) for 24 h and then washed with ultrapure water before use. For the preconcentration of Cd, firstly, 25 ml of *C. vulgaris* culture solution, 0.5 ml of 0.05 mol l$^{-1}$ Na$_2$B$_4$O$_7$-NaOH buffer solution (pH 10) and different amounts (3.5, 7, 18, 56, 108, 230, 480, 595, 812 µl and 1.5, 3.4, 6, 11 ml) of Cd$^{2+}$ stock solution were added into a 100 ml glass flask in sequence, then a certain volume of ultrapure water was added to dilute the reaction solution to 40 ml. The reaction solution was stirred at 150 r.p.m. for 1 min. Then 10 ml of the reaction solution was suction filtered on a mixed cellulose lipid microfiltration membrane with a pore size of 0.22 µm under vacuum using a sand core filter with an inner diameter of 15.1 mm. The obtained thin samples with loaded algal cells and adsorbed Cd were dried under air conditions and then directly measured by an EDXRF spectrometer. The filtrates were transferred to a 10 ml colorimetric tube and the concentrations of residual Cd$^{2+}$ in the filtrates were measured by an iCAP RQ inductively coupled plasma mass spectrometer (ICP-MS; Thermo Fisher Scientific, Germany). Under the same conditions, a series of 40 ml reference samples were prepared in the same way by mixing a certain volume of ultrapure water, 0.5 ml of 0.05 mol l$^{-1}$ Na$_2$B$_4$O$_7$-NaOH buffer solution (pH = 10) and the same volume of Cd$^{2+}$ stock solution as above (3.5, 7, 18, 56, 108, 230, 480, 595, 812 µl and 1.5, 3.4, 6, 11 ml) in 100 ml glass flasks. Cd$^{2+}$ concentrations in the reference samples were determined by ICP-MS as the initial concentrations of Cd$^{2+}$ in the reaction solutions. Then adsorption efficiency of Cd by *C. vulgaris* was calculated according to following equation (2.1):

$$P(\%) = \left[\frac{C_0 - C_t}{C_0}\right] \times 100\%. \tag{2.1}$$

where $P$ is the adsorption efficiency of Cd by *C. vulgaris*; $C_0$ is the initial concentration of Cd$^{2+}$ in the reaction solution; and $C_t$ is the concentration of residual Cd$^{2+}$ in the reaction solution when the contact time is t.

For the interference experiment of different coexisting ions on the adsorption of Cd by *C. vulgaris*, a series of reaction solutions containing 2 µg ml$^{-1}$ of Cd and different concentrations of inorganic coexisting species such as Na$^+$ (2000 µg ml$^{-1}$), K$^+$ (2000 µg ml$^{-1}$), NH$_4^+$ (1000 µg ml$^{-1}$), Ca$^{2+}$ (500 µg ml$^{-1}$), Mg$^{2+}$ (500 µg ml$^{-1}$), Fe$^{3+}$ (100 µg ml$^{-1}$), Mn$^{2+}$ (100 µg ml$^{-1}$), Zn$^{2+}$ (100 µg ml$^{-1}$), NO$_3^-$ (2000 µg ml$^{-1}$), SO$_4^{2-}$ (1000 µg ml$^{-1}$) and Cl$^-$ (1000 µg ml$^{-1}$) were treated according to the above preconcentration procedure. A blank sample was prepared in the same way as described above without adding coexisting ions. Cd sample solution (2 µg ml$^{-1}$) was as a reference sample. According to equation (2.1), the adsorption efficiency of Cd by *C. vulgaris* in the presence of different ions was calculated.

## 2.3. EDXRF spectrum measurement

The EDXRF spectra were measured using an energy dispersive X-ray fluorescence spectrometer (Amptek, USA). The excitation source was a Mini-X X-ray tube with Ag target, and the front end of the X-ray tube was equipped with a 2 mm bore diameter brass collimator and a 20 mil (0.508 mm) Al primary filter.

The detection device was a silicon drift detector (SDD). For the measurement of EDXRF spectra, the X-ray tube was operated at 40 kV and 20 μA, and the parameters of the detector were set as follows: accumulation time was 120 s, pulse shaping peak time was 6.4 μs, pulse shaping width was 0.8 μs, and gain was 47.47. A lead plate lined with red copper was used as a protective shield for X-rays. The channel number and energy value of the XRF spectrum abscissa were calibrated with iron element (Fe) K$\alpha$ characteristic peak (6.4 keV) and manganese element (Mn) K$\alpha$ characteristic peak (17.44 keV) using a 316 stainless steel plate. The channel range of the EDXRF spectrum was 1–4096, and the corresponding energy range was 0–27.20 keV.

## 2.4. Analysis of real water samples

Real water samples from Tian Er lake, Si Li river, Nan Fei river and tap water in Hefei, Anhui Province were collected in polyethylene plastic bottles which had been soaked in 10% nitric acid for more than 24 h and then cleared with ultrapure water. The water samples were first filtered through mixed cellulose lipid microfiltration membranes with a pore size of 0.45 μm and a diameter of 50 mm (Shanghai Xingya Purification Material Factory, China) and then the concentrations of Cd in those real water samples were detected by ICP-MS. After that, the real water samples were spiked with 4.9, 8.2 and 10.8 μg ml$^{-1}$ Cd$^{2+}$, respectively. The same preconcentration method as above was adopted to extract Cd$^{2+}$ from the spiked real water samples to form Cd-enriched thin samples. At the same time, Cd calibration standard samples and blank samples were prepared according to the above preconcentration procedure, and the concentrations of standard samples were determined by ICP-MS. The obtained Cd-enriched thin samples corresponding real water samples and Cd calibration standard samples were measured by the EDXRF spectrometer under the same experimental conditions. Then a recovery test was carried out on the real water samples to evaluate the reliability of the developed method.

# 3. Results and discussion

## 3.1. Characterization of *C. vulgaris* culture solution

During the cultivation process, *C. vulgaris* cells were uniformly suspended in the culture solution, and no obvious agglomeration or flake formation was observed. When *C. vulgaris* were cultured for 8 days, microscopic imaging characterization result showed that the cells of *C. vulgaris* were spherical shape with a diameter of about 5 μm and almost monodisperse. The characteristics of small cells and large specific surface area was very beneficial to the adsorption of heavy metals Cd in water. At this time, the pH value of *C. vulgaris* culture solution was 10, and the biomass of *C. vulgaris* in culture solution was 0.8253 g l$^{-1}$.

## 3.2. Optimization of adsorption conditions

In order to obtain the optimal experimental conditions for the adsorption of heavy metal Cd and realize the high-efficiency preconcentration of Cd from water samples by *C. vulgaris*, the influences of some adsorption conditions such as solution environment of *C. vulgaris* cells, contact time, pH value of reaction solution, and initial Cd concentration on the adsorption efficiency of Cd were investigated.

### 3.2.1. Effects of solution environment of algae cells and contact time

As a freshwater microalgae, due to its small cell size and large specific surface area, *C. vulgaris* is suitable for the adsorption of heavy metals in water samples. So in this study, *C. vulgaris* was selected as an adsorbent for the preconcentration of Cd from water samples prior to the determination of Cd by EDXRF spectrometry. The solution environment in which algae cells are located may affect the adsorption capacity of *C. vulgaris* for Cd. So the influences of two solution environments of *C. vulgaris* cells including ultrapure water and culture solution and contact time on the adsorption efficiency of Cd$^{2+}$ were first studied. The *C. vulgaris* ultrapure water suspension was prepared by centrifuging and washing the algal cells in culture solution and then re-dispersing them in an equal volume of ultrapure water. The same amount of *C. vulgaris* in culture solution and *C. vulgaris* in ultrapure water reacted with 4.95 μg ml$^{-1}$ of Cd$^{2+}$, respectively. The adsorption efficiencies of Cd$^{2+}$ at different contact times are shown in figure 1. From figure 1, we can see that when the contact time was 1 min, for

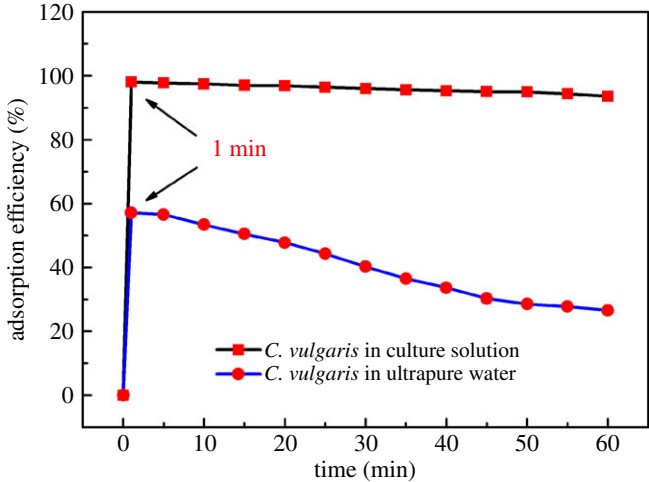

**Figure 1.** Adsorption efficiency of Cd by *C. vulgaris* in culture solution and in ultrapure water at different adsorption times.

*C. vulgaris* in both culture solution and ultrapure water, the adsorption efficiencies of $Cd^{2+}$ were the highest. This indicates that *C. vulgaris* has a fast adsorption property for $Cd^{2+}$. By comparing the adsorption characteristics of *C. vulgaris* in the two solution environments for $Cd^{2+}$, it can be seen that the adsorption efficiency of *C. vulgaris* in culture solution was much higher than that of *C. vulgaris* dispersed in ultrapure water. For *C. vulgaris* in culture solution, when the contact time was 1 min, the adsorption efficiency of Cd was up to 98.10%, while for *C. vulgaris* in ultrapure water, the adsorption efficiency of $Cd^{2+}$ was only 57.15%. When the contact time was more than 1 min, for *C. vulgaris* in both culture solution and ultrapure water, the adsorption efficiency decreased slowly with the increase of contact time, but the decreasing trend of *C. vulgaris* in ultrapure water was greater than that of *C. vulgaris* in culture solution. This indicates that $Cd^{2+}$ adsorbed on *C. vulgaris* cells had a desorption process when the contact time was more than 1 min, and in ultrapure water, the desorption rate of $Cd^{2+}$ on *C. vulgaris* cells was faster than that on *C. vulgaris* cells in culture solution. Therefore, the solution environment in which *C. vulgaris* cells were located had an important influence on the adsorption of $Cd^{2+}$. Compared with *C. vulgaris* in ultrapure water, *C. vulgaris* in culture solution had better adsorption capability for $Cd^{2+}$, which was more suitable for the rapid and efficient preconcentration of Cd from water samples. This may be because the *C. vulgaris* in the original culture solution had a good activity state, and the more active *C. vulgaris* cells themselves had strong adsorption capacity for Cd. In addition, there may be some secretions attached to the surface of *C. vulgaris* cells in the original culture solution during the cultivation process. These secretions may be more conductive to the adsorption of heavy metal Cd by *C. vulgaris* cells. However, since the activity of *C. vulgaris* cells after being centrifuged and washed would be reduced, and some secretions on the cell surface or in the culture solution were also washed or discarded away, so the adsorption capacity of *C. vulgaris* re-dispersed in ultrapure water for Cd was reduced. Therefore, in the subsequent study, *C. vulgaris* in the culture solution was used as the adsorbent for the preconcentration of Cd from *water* samples before the detection of Cd by EDXRF spectrometry, and 1 min was the optimal adsorption time.

### 3.2.2. Effect of reaction solution pH

Many research results have shown that pH is one of the important factors which influence the adsorption process of heavy metals by algae [25,26]. So in this study, the effect of pH on the adsorption capacity of *C. vulgaris* for Cd was investigated. For the mixture of *C. vulgaris* culture solution and $Cd^{2+}$, the pH value of the reaction system was adjusted from 10 to 2.56 by dropwise adding 0.1 mol l$^{-1}$ NaOH or 2% (v/v) dilute $HNO_3$ solutions. Under different pH conditions, the adsorption efficiencies of $Cd^{2+}$ by *C. vulgaris* are shown in figure 2. It can be observed from figure 2 that when the pH of reaction solution was 10, the adsorption efficiency of $Cd^{2+}$ was the highest, and the maximum adsorption efficiency was 97.95%. When the pH of reaction solution decreased from 10 to 7.5, the adsorption efficiency of $Cd^{2+}$ showed a slight reduction trend, and the adsorption efficiency decreased from 97.95% to 85.31%. However, when the pH of reaction solution continued to decrease from 7.5 to 2.56, the adsorption efficiency showed a rapid reduction trend, and the adsorption efficiency decreased from 85.31% to 0.45%. This

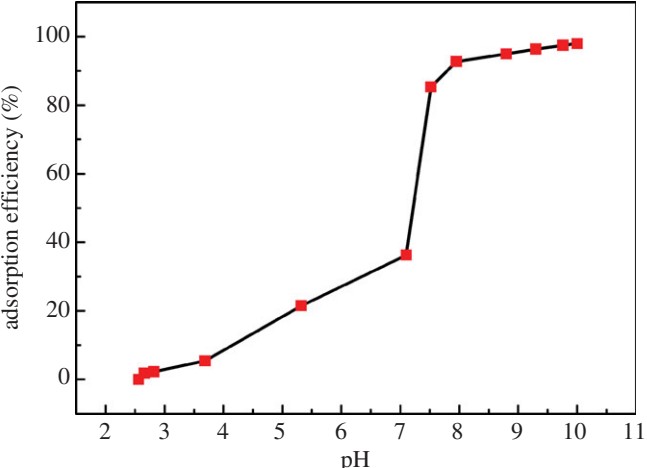

**Figure 2.** Adsorption efficiency of Cd under different pH conditions.

may be that when the pH of reaction solution was less than 7.5, there was a competitive adsorption relationship between $H^+$ and $Cd^{2+}$ in the reaction solution, and a part of the functional groups on the surface of algal cells was combined with $H^+$, so the adsorption sites participating in the adsorption of $Cd^{2+}$ were reduced. As a result, the adsorption efficiency of $Cd^{2+}$ was lower in the reaction solution with a pH of less than 7.5. The above results indicate that the alkaline solution environment with pH in the range of 7.5–10 was more favourable for the adsorption of $Cd^{2+}$ by *C. vulgaris*. When the pH value of the reaction solution system of *C. vulgaris* culture solution and Cd was 10, the adsorption capacity of *C. vulgaris* was the greatest, and the adsorption efficiency of $Cd^{2+}$ was the highest. Therefore, the optimal pH of the reaction solution was 10. In the subsequent study, for the process of adsorption and preconcentration of Cd from water samples, the reaction solution was adjusted to 10 by adding 0.5 ml of $0.05 \, mol \, l^{-1} \, Na_2B_4O_7$-NaOH buffer solution with a pH of 10.

### 3.2.3. Effect of initial Cd concentration

In this study, the high-efficiency preconcentration of Cd from water samples was an important precondition for the detection of Cd by EDXRF spectrometry. In order to determine the adsorption capacity of *C. vulgaris* for different concentrations of Cd, when *C. vulgaris* reacted with $Cd^{2+}$ for 1 min, the effect of different initial $Cd^{2+}$ concentrations in the reaction solution on the adsorption efficiency of $Cd^{2+}$ was investigated, and the result is shown in figure 3. It can be seen from figure 3 that when the initial concentration of $Cd^{2+}$ was less than $0.703 \, \mu g \, ml^{-1}$, the adsorption efficiency of $Cd^{2+}$ was less than 90%, but the adsorption efficiency increased with the increase of $Cd^{2+}$ concentration in this range. When the initial concentration of $Cd^{2+}$ was in the range of $0.703$–$74.957 \, \mu g \, ml^{-1}$, all the adsorption efficiencies were more than 90%. Among them, for the initial $Cd^{2+}$ concentrations of 18.808 and $42.314 \, \mu g \, ml^{-1}$, the adsorption efficiencies of $Cd^{2+}$ were as high as 99.04% and 99.31%, respectively. However, when the initial concentration of $Cd^{2+}$ was greater than $74.957 \, \mu g \, ml^{-1}$, the adsorption efficiency decreased as the concentration of $Cd^{2+}$ increased. For example, for a reaction solution with an initial $Cd^{2+}$ concentration of $137.866 \, \mu g \, ml^{-1}$, the adsorption efficiency had reduced to 65.62%. This may be because when the initial concentration of $Cd^{2+}$ was greater than $74.957 \, \mu g \, ml^{-1}$, the functional groups participating in $Cd^{2+}$ adsorption on the surface of *C. vulgaris* cells were all combined with $Cd^{2+}$, and the adsorption points on the algal cells had reached saturation. Therefore, when the concentration of $Cd^{2+}$ continued to increase, the adsorption amount of $Cd^{2+}$ by *C. vulgaris* didn't increase any more, so the calculated adsorption efficiency of $Cd^{2+}$ decreased. As a result, in order to realize the effective preconcentration of Cd from water samples, for the reaction solution of *C. vulgaris* culture solution and Cd, the optimal initial concentration range of Cd was $0.703$–$74.957 \, \mu g \, ml^{-1}$.

### 3.3. Influence of coexisting ions

Due to the existence of a variety of inorganic ions in real environmental water samples, these ions may interfere with the adsorption of heavy metal Cd by *C. vulgaris*. Therefore, the effect of different coexisting ions on the adsorption efficiency of Cd by *C. vulgaris* was investigated. For this purpose, a series of

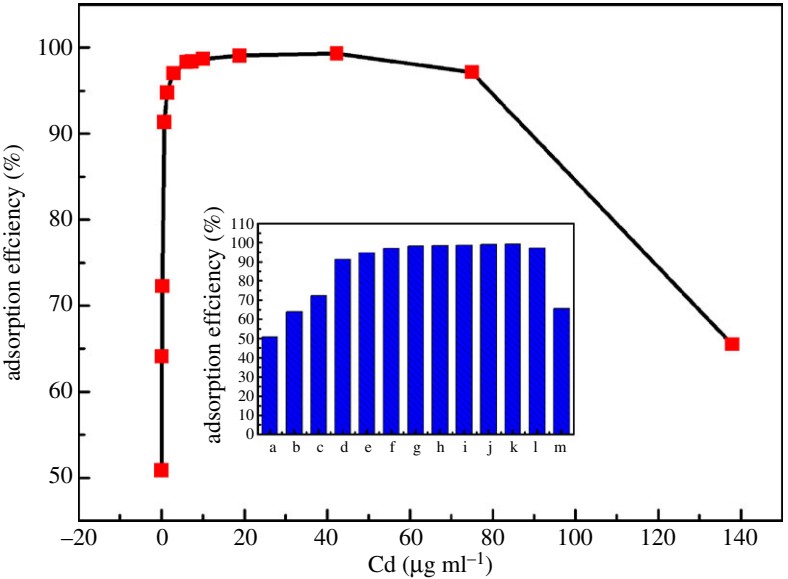

**Figure 3.** The change of adsorption efficiency of Cd with initial Cd concentration (a: 0.044 µg ml⁻¹, b: 0.090 µg ml⁻¹, c: 0.229 µg ml⁻¹, d: 0.703 µg ml⁻¹, e: 1.352 µg ml⁻¹, f: 2.876 µg ml⁻¹, g: 6.014 µg ml⁻¹, h: 7.431 µg ml⁻¹, i: 10.152 µg ml⁻¹, j: 18.808 µg ml⁻¹, k: 42.314 µg ml⁻¹, l: 74.957 µg ml⁻¹, m: 137.866 µg ml⁻¹).

**Table 1.** Adsorption efficiency of Cd (2 µg ml⁻¹) by *C. vulgaris* in the presence of coexisting ions.

| coexisting ions | concentration (µg ml⁻¹) | adsorption efficiency (%) | change (%) |
|---|---|---|---|
| blank | | 95.53 ± 2.51 | |
| $Na^+$ | 2000 | 92.46 ± 3.47 | 3.21 |
| $K^+$ | 2000 | 92.98 ± 3.21 | 2.67 |
| $NH_4^+$ | 1000 | 91.97 ± 3.76 | 3.73 |
| $Ca^{2+}$ | 500 | 93.44 ± 2.90 | 2.19 |
| $Mg^{2+}$ | 500 | 92.75 ± 3.05 | 2.91 |
| $Fe^{3+}$ | 100 | 91.69 ± 2.88 | 4.02 |
| $Mn^{2+}$ | 100 | 92.79 ± 2.66 | 2.87 |
| $Zn^{2+}$ | 100 | 91.41 ± 2.47 | 4.31 |
| $NO_3^-$ | 2000 | 93.28 ± 3.07 | 2.36 |
| $SO_4^{2-}$ | 1000 | 92.98 ± 2.75 | 2.67 |
| $Cl^-$ | 1000 | 92.66 ± 2.55 | 3.00 |

reaction solutions containing 2 µg ml⁻¹ of Cd and different concentrations of inorganic coexisting species such as $Na^+$, $K^+$, $NH_4^+$, $Ca^{2+}$, $Mg^{2+}$, $Fe^{3+}$, $Mn^{2+}$, $Zn^{2+}$, $NO_3^-$, $SO_4^{2-}$ and $Cl^-$ were treated according to the proposed preconcentration procedure under the optimum experimental conditions, and the adsorption efficiencies of Cd by *C. vulgaris* are shown in table 1. As can be seen from table 1, compared with the blank sample, when 1000 fold $Na^+$, $K^+$ and $NO_3^-$, 500 fold $NH_4^+$, $SO_4^{2-}$ and $Cl^-$, 250 fold $Ca^{2+}$ and $Mg^{2+}$, and 50 fold $Fe^{3+}$, $Mn^{2+}$ and $Zn^{2+}$ coexisted with 2 µg ml⁻¹ Cd, all the changes of adsorption efficiency of Cd by *C. vulgaris* were within 5%. These results indicated that even at high concentrations, those coexisting inorganic ions had no significant influence on the adsorption of Cd by *C. vulgaris*.

## 3.4. Uniformity of obtained Cd-enriched thin samples

Under the above optimal adsorption experimental conditions, for the reaction solutions with initial Cd concentration of 0.703, 1.352, 2.876, 6.014, 7.431, 10.152, 18.808, 42.314 and 74.957 µg ml⁻¹, mixed

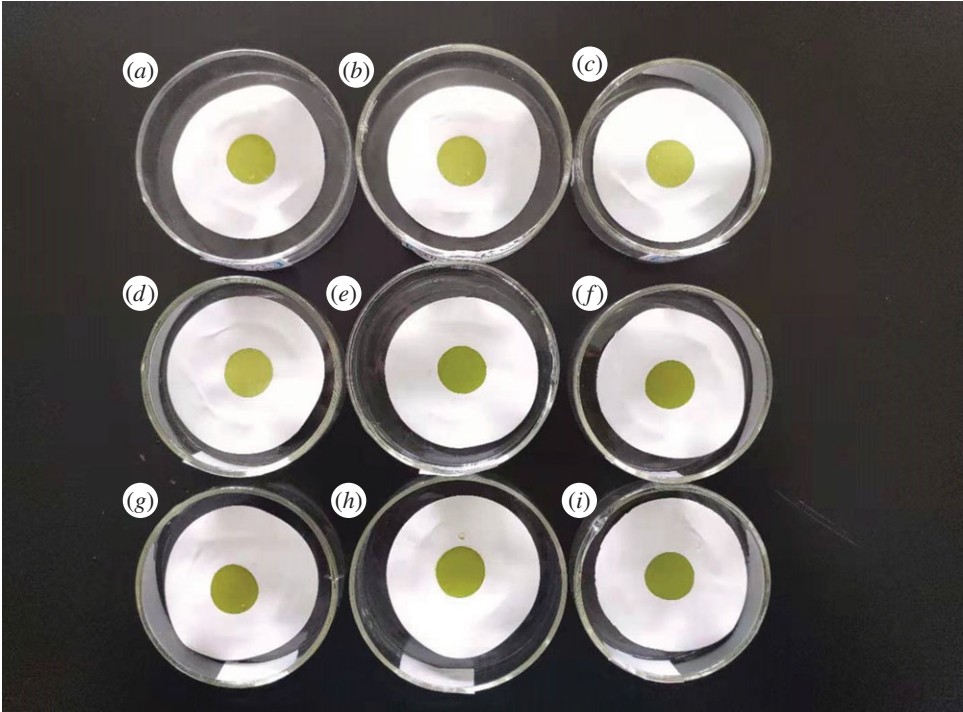

**Figure 4.** The formed Cd-enriched thin samples corresponding to different initial Cd concentrations (a: 0.703 µg ml$^{-1}$, b: 1.352 µg ml$^{-1}$, c: 2.876 µg ml$^{-1}$, d: 6.014 µg ml$^{-1}$, e: 7.431 µg ml$^{-1}$, f: 10.152 µg ml$^{-1}$, g: 18.808 µg ml$^{-1}$, h: 42.314 µg ml$^{-1}$, i: 74.957 µg ml$^{-1}$).

cellulose lipid microfiltration membranes with a pore size of 0.22 µm were used as the supporting membranes, and 10 ml of the reaction solutions were suction filtered to obtain Cd-enriched thin samples. The formed Cd thin samples corresponding to different initial Cd concentrations are shown in figure 4. The diameter of the circular Cd-enriched region on the Cd thin samples was 15.1 mm, which was much larger than the inner diameter of the exit port of the collimator at the front end of the X-ray tube in the EDXRF spectrometer. The uniformity of Cd-enriched thin samples was the key for the accurate and stable detection of Cd in water samples by EDXRF spectrometry. In order to test the uniformity of the obtained Cd-enriched thin samples prepared by the preconcentration method, EDXRF spectrum measurement was performed at six different positions in the Cd-enriched region for each Cd thin sample. Then for the six measurement results, the relative standard deviation (RSD) of the gross integrated fluorescence intensity of Cd K$\alpha$ characteristic peak was calculated, and the results are shown in figure 5. It can be seen from figure 5 that for the obtained Cd thin samples corresponding to the initial Cd concentration in the range of 0.703–74.957 µg ml$^{-1}$, all the RSD values of the EDXRF spectrum measurement results at six different positions in the Cd-enriched region were no more than 2.22%. These results indicate that the Cd-enriched thin samples obtained by the preconcentration method had very good uniformity and could be used as test samples to be directly measured by XRF spectrometer.

## 3.5. Detection of Cd by EDXRF

For the obtained each Cd-enriched thin sample corresponding to initial Cd concentration in the range of 0.703–74.957 µg ml$^{-1}$ by the above preconcentration method, EDXRF spectrum measurement was performed six times, and the average spectrum of the measured six EDXRF spectra was obtained. The average EDXRF spectra near Cd K$\alpha$ characteristic peak (23.11 keV) corresponding to different initial Cd concentrations are shown in figure 6. From figure 6, it can be seen that as the concentration of Cd increased, the intensity of Cd K$\alpha$ characteristic peak gradually increased. For Cd K$\alpha$ characteristic peak in the EDXRF spectrum of each Cd thin sample, the net integrated fluorescence intensity was calculated by the gross integral fluorescence intensity of the measured EDXRF spectrum being subtracted from the background integral fluorescence intensity of the blank filter membrane sample loaded *C. valgaris* cells. The average value of the net integral fluorescence intensity of Cd K$\alpha$ characteristic peak was linearly fitted with the initial concentration of Cd, and the result is shown in figure 7. As can been seen from figure 7, when the initial concentration of Cd was in the range of 0.703–74.957 µg ml$^{-1}$, the net integral fluorescence

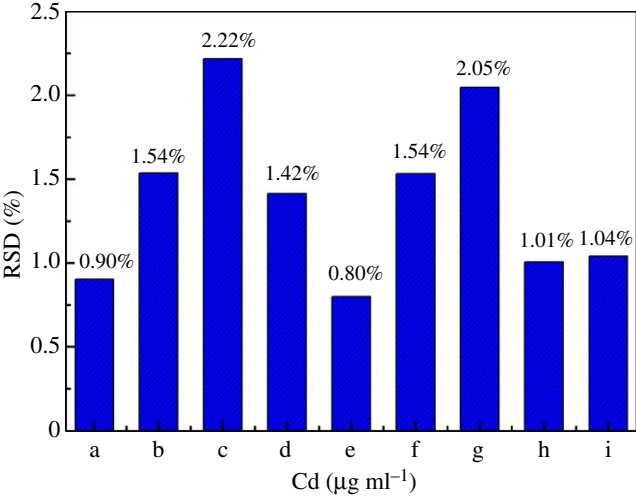

**Figure 5.** The uniformity of Cd thin samples corresponding to different initial Cd concentration (a: 0.703 µg ml⁻¹, b: 1.352 µg ml⁻¹, c: 2.876 µg ml⁻¹, d: 6.014 µg ml⁻¹, e: 7.431 µg ml⁻¹, f: 10.152 µg ml⁻¹, g: 18.808 µg ml⁻¹, h: 42.314 µg ml⁻¹, i: 74.957 µg ml⁻¹).

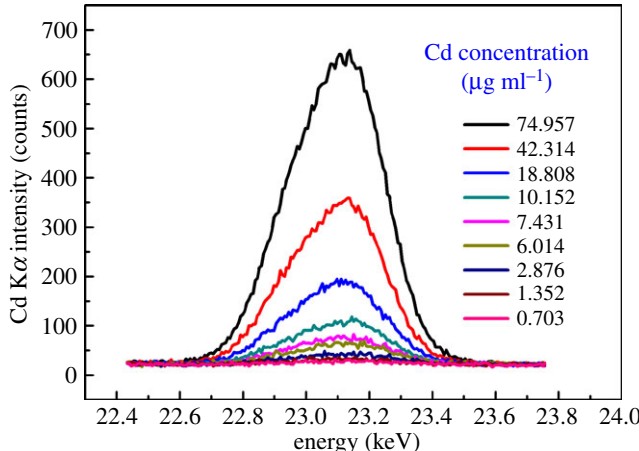

**Figure 6.** K$\alpha$ characteristics EDXRF spectra of Cd corresponding to different initial concentrations.

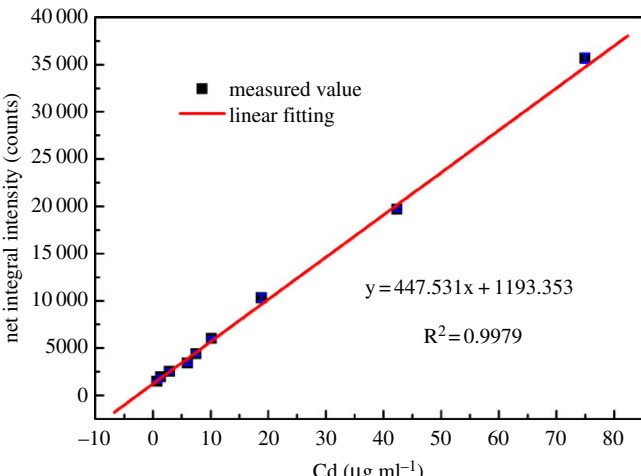

**Figure 7.** Calibration curve between the net integral fluorescence intensity of Cd K$\alpha$ characteristic peak and Cd concentration.

**Table 2.** Comparison of some reported methods based on XRF spectrometry for the detection of Cd in water samples.

| adsorbent | adsorption time (min) | technique | LOD ($\mu$g l$^{-1}$) (sample volume, counting time) | ref. |
|---|---|---|---|---|
| thin layers activated with the extractant Aliquat 336 | 150 | HE-P-EDXRF (Gd target, 100 kV, 6 mA) | 0.7 (190 ml, 1000 s) | [27] |
| Dowex 50 cation-exchange resin | 67 | EDXRF (W target, 50 kV, 0.8 mA) | 3 (100 ml, 700 s) | [28] |
| dithizone functionalized grapheneas | 10 | WDXRF (Rh target, 60 kV, 60 mA) | 6.1 (100 ml, 300 s) | [29] |
| nano-TiO$_2$/cellulose paper composite | 180 | EDXRF (Ag target, -, -) | 0.51 (1000 ml, -) | [13] |
| iminodiacetate extraction disk | 10 | WDXRF (Rh target, 50 kV, 80 mA) | 7 (1000 ml, 60 s) | [30] |
| CoFe$_2$O$_4$ nanoparticles impregnated with 1-(2-pyridylazo)-naphthol | 10 | EDXRF (Pd target, 40 kV, 30 mA) | 13 (25 ml, 90 s) | [31] |
| activated membranes by anionic exchanger Aliquat 336 | 150 | WDXRF (Rh target, 50 kV, 20 mA) | 170 (200 ml, 6 s) | [32] |
| multiwalled carbon nanotubes | 5 | TXRF (W target, 50 kV, 1 mA) | 1 (20 ml, 2000 s) | [33] |
| freshwater green algae *Chlorella vulgaris* | 1 | EDXRF (Ag target, 40 kV, 20 $\mu$A) | 65.4 (10 ml, 120 s) | this work |

intensity of Cd K$\alpha$ characteristic peak had a very good linear relationship with the initial concentration of Cd, and the linear calibration plot was $F = 447.53061C + 1193.35273$ with the linear correlation coefficient $R^2$ of 0.9979, where $F$ is the net integral fluorescence intensity of Cd K$\alpha$ characteristic peak, and C is the initial concentration of Cd (the unit is $\mu$g ml$^{-1}$). The Cd$^{2+}$ standard solution with an initial concentration of 5.133 $\mu$g ml$^{-1}$ was repeatedly detected for six times, and the RSD of the six times detection results was 3.84%, which was within 5%. For the blank filter membrane sample loaded *C. valgaris* cells, the EDXRF spectrum measurement was repeated 11 times under the same experimental conditions, the standard deviation ($\sigma$) of the integrated fluorescence intensity obtained from 11 repeated measurements was 9.7617, and the slope ($k$) of the above linear calibration plot was 447.503061 ml $\mu$g$^{-1}$. So according to the $3\sigma/k$ IUPAC criteria ($\sigma$ is the standard deviation of multiple measurements for blank sample; $k$ is the slope of linear calibration plot), for the formed Cd thin samples with a Cd-enriched region of 15.1 mm in diameter obtained by the developed preconcentration method with suction filtration of 10 ml reaction solution, the detection limit of this proposed EDXRF detection method was 0.0654 $\mu$g ml$^{-1}$, which was lower than the maximum allowable discharge concentration of Cd specified in discharge standards of water pollutants for various industries of China (nos. GB 13456–2012, GB 25463–2010 and GB 21904–2008). So this method can be applied to the rapid detection and early warning of excessive Cd in discharged industrial wastewaters. In addition, the accuracy of the proposed method for detection of Cd was evaluated by analysing two certified reference water samples, GBW(E)082822a-1 (certified Cd concentration: 100 $\mu$g ml$^{-1}$) and

**Table 3.** Recovery for the detection of Cd in real water samples.

| type of sample | added ($\mu$g ml$^{-1}$) | found ($\mu$g ml$^{-1}$) | RSD (%) ($n = 3$) | recovery (%) |
|---|---|---|---|---|
| Tian Er lake | 4.90 | 4.645 | 4.55 | 94.80 |
| (0.254 $\mu$g l$^{-1}$) | 8.20 | 9.518 | 3.63 | 116.07 |
| | 10.80 | 12.293 | 3.58 | 113.82 |
| Si Li river | 4.90 | 5.307 | 4.53 | 108.30 |
| (0.136 $\mu$g l$^{-1}$) | 8.20 | 9.110 | 3.02 | 111.10 |
| | 10.80 | 12.633 | 3.24 | 116.94 |
| Nan Fei river | 4.90 | 4.747 | 4.21 | 96.87 |
| (0.156 $\mu$g l$^{-1}$) | 8.20 | 9.133 | 3.35 | 111.38 |
| | 10.80 | 12.125 | 2.87 | 112.27 |
| tap water | 4.90 | 5.533 | 5.05 | 112.93 |
| | 8.20 | 8.904 | 4.69 | 108.56 |
| | 10.80 | 11.980 | 3.49 | 110.93 |

GBW(E)082822b-2 (certified Cd concentration: 500 $\mu$g ml$^{-1}$). Prior to preconcentration, the reaction solution composed of 25 ml of *C. vulgaris* culture solution, 0.5 ml of Na$_2$B$_4$O$_7$–NaOH buffer solution (pH 10) and 1 ml Cd certified reference water sample was diluted to 40 ml with ultrapure water. Then preconcentration and detection of Cd were performed according to the proposed method. Compared with the certified Cd concentrations of the two reference water samples, the relative errors of the detection results were 4.7% and 3.1%, respectively. Moreover, the preconcentration process and detection method of Cd in this work was compared with some reported methods based on XRF spectrometry for the detection of Cd in water samples. As can be seen from table 2, for the preconcentration process, the adsorbents used in the reported works all needed to be prepared or pretreated. The preparation and pretreatment processes were very complicated and time consuming. The adsorption times of Cd by these adsorbents were usually in the range of 5–180 min. Compared with these adsorbents, in this work, the adsorbent of *C. vulgaris* in culture solution could be directly used to adsorb Cd without any complicated pretreatment processes, and it also had the advantages of easy to obtain, easy to culture, low cost, and environmental friendliness. In addition, the adsorption time of *C. vulgaris* to Cd was only 1 min, which was much shorter than the adsorption time of other adsorbents used in the published works to Cd. For the detection method of Cd, the detection limit was related to both the sample volume used in the preconcentration process and the accumulation time in the XRF spectrum measurement process. Generally, the more the sample volume was, the longer the accumulation time was, the lower the detection limit of method was. For the detection method of Cd proposed in this work, if the volume of the reaction solution used for suction filtration was increased, the diameter of Cd-enriched region on Cd thin samples was decreased, or the accumulation time in the EDXRF spectrum measurement process was increased, the detection limit of this method would be further improved and decreased.

## 3.6. Application to analysis of real water samples

The potential applicability of the proposed method for Cd detection in real water samples was further investigated by adopting standard addition method. The surface lake water and river water samples were collected from Tian Er lake, Si Li river, Nan Fei river. The tap water sample was collected from the faucet outlet in our laboratory. All real water samples were stored in precleaned polyethylene plastic bottles and filtered through 0.45 $\mu$m mixed cellulose lipid microfiltration membranes before use. Then the concentration of Cd in the real water samples was detected by ICP-MS, and the concentrations of Cd in Tian Er lake, Si Li river, Nan Fei river were 0.254, 0.136 and 0.156 $\mu$g l$^{-1}$, respectively. However, Cd was not found in tap water by ICP-MS. After that, a recovery study was carried out on the real water samples by being spiked with 4.90, 8.20 and 10.80 $\mu$g ml$^{-1}$ Cd$^{2+}$ to evaluate the developed method. The corresponding results obtained by the same preconcentration procedure and EDXRF spectral measurement method described above are listed in table 3. As shown in table 3, the recoveries of Cd in

the real water samples ranged from 94.80% to 116.94%. For the spiked water samples, all the RSD values of the three repeated measurements were no more than 5.05%. These results indicate that the developed method by the combination of preconcentration with *C. vulgaris* and EDXRF spectrometry could be well applied to the detection of Cd in real water.

## 4. Conclusion

In summary, since *C. vulgaris* in culture solution had a high adsorption capacity for Cd, and the maximum adsorption efficiency could be obtained when the contact time was 1 min and the pH of reaction solution was 10, a simple, rapid, economical and environmentally friendly method for the detection of heavy metal Cd in water samples was developed based on preconcentration with *C. vulgaris* combined with EDXRF spectrometry in this study. For the preconcentration process, *C. vulgaris* as an adsorbent was easy to obtain, low cost and no additional pretreatment was required before use. After preconcentration, the obtained Cd-enriched thin samples had very good uniformity, which could be directly measured by EDXRF spectrometry, and the net integral fluorescence intensity of Cd K$\alpha$ characteristic peak had a very good linear relationship with the initial concentration of Cd$^{2+}$ in the range of 0.703–74.957 µg ml$^{-1}$. When the Cd thin samples with a Cd-enriched region of 15.1 mm in diameter were formed by the developed preconcentration method with suction filtration of 10 ml reaction solution, the detection limit of this method was 0.0654 µg ml$^{-1}$, which was lower than the maximum allowable discharge concentration of Cd specified in discharge standards of water pollutants for various industries of China. Moreover, the proposed method was simple to operate, and could effectively remove the influence of matrix effect of water samples and effectively improve the sensitivity and stability of XRF spectrometry, directly detecting heavy metals in water samples, which was very suitable for the real-time, rapid and online detection and early warning of excessive Cd in discharged industrial wastewaters. This work will provide a methodological basis for the development of real-time, rapid and online monitoring technology and instrument of heavy metal pollutants in water. In our future studies, we will further optimize this method in order to reduce the detection limit and improve the sensitivity. It is hoped that our future research results can provide methods for the detection and warning of excessive Cd in other waters such as drinking water and surface water.

Data accessibility. All relevant data of our research are accessible from Dryad Digital Repository: https://doi.org/10.5061/dryad.6hdr7sqwv [34]. (https://datadryad.org/stash/share/JCu5SUypEJ7l5CArrMq3ne0LsxTexArQIPP2vc117U4).
Authors' contributions. T.G. carried out the design of the study, the experimental research work and data analysis, drafted and wrote the manuscript. N.Z. critically revised the manuscript. G.Y. participated in data analysis and helped draft the manuscript. M.C. cultured algae samples. X.W. and H.H. participated in the experimental research work and collected field data. All authors gave final approval for publication and agree to be held accountable for the work performed therein.
Competing interests. We declare we have no competing interests.
Funding. This work was supported by the National Natural Science Foundation of China (grant no. 61805254); the National Key Research and Development Program of China (grant no. 2016YFC1400602); the Chinese Academy of Sciences Instrument and Equipment Function Development Technology Innovation Project (grant no. Y93H3g1251); the Anhui Provincial Excellent Youth Science Foundation of China (grant no. 1908085J23); and the Marine National Laboratory Open Foundation Program of China (grant no. QNLM2016ORP0312).
Acknowledgements. We are grateful Prof. Wenqing Liu, who provided guidance for improving our paper.

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
