## [Reviewer comments · Royal Society Open Science]

Review History

RSOS-200182.R0 (Original submission)

Review form: Reviewer 1

Is the manuscript scientifically sound in its present form?

Yes

Are the interpretations and conclusions justified by the results?

Yes

Is the language acceptable?

Yes

Do you have any ethical concerns with this paper?

No

Have you any concerns about statistical analyses in this paper?

No

Recommendation?

Accept with minor revision (please list in comments)

Comments to the Author(s)

This paper is an interesting study. The data support the conclusion, and I am very interested to read the paper.

Review form: Reviewer 2**Is the manuscript scientifically sound in its present form?**

No

Are the interpretations and conclusions justified by the results?

No

Is the language acceptable?

Yes

Do you have any ethical concerns with this paper?

No

Have you any concerns about statistical analyses in this paper?

No

Recommendation?

Major revision is needed (please make suggestions in comments)

Comments to the Author(s)

The manuscript presents a preconcentration procedure with *Chlorella vulgaris* for posterior determination of Cd in water samples using energy dispersive X-ray fluorescence spectrometry. The subject is interesting and important and has merit for publication. There are comments/addings, and perhaps additional experiments, which could be considered prior to publication:

Experimental

- The *C. vulgaris* culture solution need to be characterized. The mass of adsorbent in the suspension need to be determined.
- Please add more information about the mixed cellulose membranes.
- Preparation of the calibration standards and blank need to be described.
- The sample volume needs to be informed in the procedure.

Results and discussion

- Discuss why the use of the *C. vulgaris* in culture solution was more efficient.
- The effect of pH is not clear. How was the pH control performed in the system? For example, the authors affirm that the method can applied in discharged industrial wastewaters. However, the final pH of the system can be affected by the sample acidity.
- Please present the study of the effect of matrix.
- The *C. vulgaris* could adsorb other inorganic species from water. In this way, a study of potential interferences need to be presented.
- Inform how the detection limit was calculated.
- The method was unable to determine cadmium in the water samples without adding the analyte. However, the detection limit of the method can be improved and decreased by varying, for example, the sample volume. Additional experiments need to be present with this intention.
- The method is not sufficiently validated. For example: the results obtained need to be compared with a comparative method or a water certified reference material needs to be analyzed.

Decision letter (RSOS-200182.R0)

06-Mar-2020

Dear Dr Gan:

Title: Preconcentration with *Chlorella vulgaris* combined with energy dispersive X-ray fluorescence spectrometry for rapid determination of Cd in water
Manuscript ID: RSOS-200182

The editor assigned to your manuscript has now received comments from reviewers. We would like you to revise your paper in accordance with the referee and Subject Editor suggestions which can be found below (not including confidential reports to the Editor). Please note this decision does not guarantee eventual acceptance.

Please submit your revised paper before 29-Mar-2020. Please note that the revision deadline will expire at 00.00am on this date. If we do not hear from you within this time then it will be assumed that the paper has been withdrawn. In exceptional circumstances, extensions may be possible if agreed with the Editorial Office in advance. We do not allow multiple rounds of revision so we urge you to make every effort to fully address all of the comments at this stage. If deemed necessary by the Editors, your manuscript will be sent back to one or more of the original reviewers for assessment. If the original reviewers are not available we may invite new reviewers.

RSC Associate Editor:
Comments to the Author:
(There are no comments.)

RSC Subject Editor:
Comments to the Author:
(There are no comments.)

Reviewers' Comments to Author:
Reviewer: 1

Comments to the Author(s)
This paper is an interesting study. The data support the conclusion, and I am very interested to read the paper.

Reviewer: 2

Comments to the Author(s)
The manuscript presents a preconcentration procedure with *Chlorella vulgaris* for posterior determination of Cd in water samples using energy dispersive X-ray fluorescence spectrometry. The subject is interesting and important and has merit for publication. There are comments/addings, and perhaps additional experiments, which could be considered prior to publication:

Experimental

- The *C. vulgaris* culture solution need to be characterized. The mass of adsorbent in the suspension need to be determined.
- Please add more information about the mixed cellulose membranes.
- Preparation of the calibration standards and blank need to be described.
- The sample volume needs to be informed in the procedure.

Results and discussion

- Discuss why the use of the *C. vulgaris* in culture solution was more efficient.
- The effect of pH is not clear. How was the pH control performed in the system? For example, the authors affirm that the method can applied in discharged industrial wastewaters. However, the final pH of the system can be affected by the sample acidity.
- Please present the study of the effect of matrix.
- The *C. vulgaris* could adsorb other inorganic species from water. In this way, a study of potential interferents need to be presented.
- Inform how the detection limit was calculated.
- The method was unable to determine cadmium in the water samples without adding the analyte. However, the detection limit of the method can be improved and decreased by varying, for example, the sample volume. Additional experiments need to be present with this intention.
- The method is not sufficiently validated. For example: the results obtained need to be compared with a comparative method or a water certified reference material needs to be analyzed.

Author's Response to Decision Letter for (RSOS-200182.R0)

See Appendix A.

Decision letter (RSOS-200182.R1)

30-Mar-2020

Dear Dr Gan:

Title: Preconcentration with *Chlorella vulgaris* combined with energy dispersive X-ray fluorescence spectrometry for rapid determination of Cd in water
Manuscript ID: RSOS-200182.R1

It is a pleasure to accept your manuscript in its current form for publication in Royal Society Open Science. The chemistry content of Royal Society Open Science is published in collaboration with the Royal Society of Chemistry.

RSC Associate Editor
Comments to the Author:
(There are no comments.)

Reviewer(s)' Comments to Author:

Appendix A

Dear Editor and Reviewers:

Thank you very much for your letter and the comments from editor and reviewers about our manuscript entitled "**Preconcentration with *Chlorella vulgaris* combined with energy dispersive X-ray fluorescence spectrometry for rapid determination of Cd in water**" (Manuscript ID: **RSOS-200182**). Those comments are all valuable and very helpful for revising and improving our paper, as well as the important guiding significance to our researches. We have considered every comment from the reviewers very carefully and have made correction exactly according to the comments. And the point by point responses to the comments made by the reviewers are listed below this letter. The changes we have made to the original manuscript were listed in the last section of this letter. We have try our best to improve the manuscript and hope meet with approval.

We appreciate for Editor and Reviewers' warm work earnestly. We would like to resubmit this revised manuscript to *Royal Society Open Science*, and hope it is acceptable for publication in the journal. In what follows, we respond to these comments and we list the changes to the original manuscript.

Once again, thank you very much for your comments and suggestions. Looking forward to hearing from you soon.

Yours sincerely,

Tingting Gan

Anhui Institute of Optics and Fine Mechanics, Chinese Academy of Sciences

Responses to the comments made by the Reviewers

Reviewer 1:

Comment: This paper is an interesting study. The data support the conclusion, and I am very interested to read the paper.

Response: Thank you very much for your interest and carefully reading our manuscript, and giving us a positive comment on this paper. And we have further revised and improved our manuscript during this revision stage in order to make it better and more acceptable for this journal. Once again, special thanks to you for your comment.

Reviewer 2:

Comments: The manuscript presents a preconcentration procedure with *Chlorella vulgaris* for posterior determination of Cd in water samples using energy dispersive X-ray fluorescence spectrometry. The subject is interesting and important and has merit for publication. There are comments/ additions, and perhaps additional experiments, which could be considered prior to publication:

Experimental

1. Comment: The *C. vulgaris* culture solution need to be characterized. The mass of adsorbent in the suspension need to be determined.

Response: Thank you very much for your valuable comments and suggestions. It is really true that in our original manuscript, the experiments of characterizing *C.*

vulgaris culture solution and determination of mass of adsorbent in the suspension were not described. And in this paper, the characterization of *C. vulgaris* culture solution and the determination of mass of adsorbent in the suspension were the premise and basis for carrying out the experimental studies on the adsorption and preconcentration of Cd in water samples by *C. vulgaris*. Therefore, although we did not describe the characterization results of *C. vulgaris* culture solution and the determination result of mass of adsorbent in the suspension, we have performed these two experiments before carrying out the experimental studies on the adsorption and preconcentration of Cd in water samples by *C. vulgaris*. But we are very sorry for our negligence of those description in our original manuscript. So according to the comments of reviewer, in our revised manuscript, we have corrected the subheading "**2.1 Algal culture**" in our original manuscript as "**2.1 Algae culture and characterization**", and at the end of this section, we have supplemented the descriptions of the experiments for characterizing *C. vulgaris* culture solution and determination of mass of adsorbent in the suspension. Moreover, in the section "**Results and discussion**", we have added a subheading "**3.1 Characterization of C. vulgaris culture solution**", and the characterization results of *C. vulgaris* culture solution and the determination results of mass of adsorbent in the suspension have been supplemented in this section.

2. Comment: Please add more information about the mixed cellulose membranes.

Response: Thank you very much for your suggestion. In our original manuscript, the information about the mixed cellulose membranes was indeed too simple and not adequate. So according to your suggestion, in our revised manuscript, we have added more detailed information about the mixed cellulose membranes in the section "**2. Experimental**" such as "*a mixed cellulose lipid microfiltration membrane with a pore size of 0.22 μm and a diameter of 50 mm (Shanghai Xingya Purification Material Factory, China)*" in line 20-21 on page 6 and "*mixed cellulose lipid microfiltration membranes with a pore size of 0.45 μm and a diameter of 50 mm*

(Shanghai Xingya Purification Material Factory, China)" in line 8-10 on page 9.

3. Comment: Preparation of the calibration standards and blank need to be described.

Response: Thank you very much for your suggestion. We are very sorry that the preparation of the calibration standards and blank was not described in our original manuscript. So according to this suggestion, in our revised manuscript, we have supplemented the detailed description of preparation of the calibration standards and blank such as "*At the same time, Cd calibration standard samples and blank samples were prepared according to the above preconcentration procedure, and the concentrations of standard samples were determined by ICP-MS.*" in line 10-12 in the section "**2.4 Analysis of real water samples**".

4. Comment: The sample volume needs to be informed in the procedure.

Response: Thank you very much for your suggestion. In our original manuscript, it is really true that we didn't explicitly describe the volume of some samples in the experimental part. So according to your suggestion, in our revised manuscript, we have clearly stated the specific sample volume such as "*.....15 mL of C. vulgaris culture solution.....*" in line 24 in section "**2.1 Algae culture and characterization**", "*For the preconcentration of Cd, firstly, 25 mL of C. vulgaris culture solution, 0.5 mL of 0.05 mol L⁻¹ Na₂B₄O₇-NaOH buffer solution (pH 10) and different amounts (3.5, 7, 18, 56, 108, 230, 480, 595, 812 μL and 1.5, 3.4, 6, 11 mL) of Cd²⁺ stock solution were added into a 100 mL glass flask in sequence, then a certain volume of ultrapure water was added to dilute the reacton solution to 40 mL.*" in line 14-18, "*Then 10 mL of the reaction solution was suction filtered on*" in line 19-20, and "*Under the same conditions, a series of 40 mL reference samples were prepared in the same way by mixing a certain volume of ultrapure water, 0.5 mL of 0.05 mol L⁻¹ Na₂B₄O₇-NaOH buffer solution (pH=10) and the same volume of Cd²⁺ stock solution as above (3.5, 7, 18, 56, 108, 230, 480, 595, 812 μL and 1.5, 3.4, 6, 11 mL)*"

in 100 mL glass flasks." in line 26-31 in the first paragraph of section "2.2 Preconcentration procedure".

Results and discussion

1. Comment: Discuss why the use of the *C. vulgaris* in culture solution was more efficient.

Response: Thank you very much for your comment. we are very sorry that in our original manuscript, we didn't discuss why the use of the *C. vulgaris* in culture solution was more effecient. So according to your comment, in our revised manuscript, we have supplemented the discussion on why the use of the *C. vulgaris* in culture solution was more efficient such as "*This may be that the C. vulgaris in the original culture solution had good activity state, and the more active C. vulgaris cells themselves had strong adsorption capacity for Cd. In addition, there may be some secretions attached to the surface of C. vulgaris cells in the original culture solution during the cultivation process. And these secretions may be more conductive to the adsorption of heavy metal Cd by C. vulgaris cells. However, since the activity of C. vulgaris cells after being centrifuged and washed would be reduced, and some secretions on the cell surface or in the culture solution were also washed or discarded away, so the adsorption capacity of C. vulgaris re-dispersed in ultrapure water for Cd was reduced.*" in line 32-41 in the section "3.2.1 Effects of solution environment of algae cells and contact time".

2. Comment: The effect of pH is not clear. How was the pH control performed in the system? For example, the authors affirm that the method can applied in discharged industrial wastewaters. However, the final pH of the system can be affected by the sample acidity.

Response: Thank you very much for your comment. We are very sorry that in our original manuscript the effect of pH and the pH control of reaction solution system

were not described clearly. So according to your comment, in our revised manuscript, the sentences in line 7-23 have been carefully modified so that we can more clearly describe how pH affected the adsorption of Cd by *C. vulgaris*. And in order to clearly describe how to control the pH of the reaction solution, we have supplemented the sentences "*For the mixture of C. vulgaris culture solution and Cd²⁺, the pH value of the reaction system was adjusted from 10 to 2.56 by dropwise adding 0.1 mol L⁻¹ NaOH or 2% (v/v) dilute HNO₃ solutions.*" in line 3-6, and "*And in the subsequent study, for the process of adsorption and preconcentration of Cd from water samples, the reaction solution was adjusted to 10 by adding 0.5 mL of 0.05 mol L⁻¹ Na₂B₄O₇-NaOH buffer solution with a pH of 10.*" in line 24-27 in section "**3.2.2 Effect of reaction solution pH**". In addition, because when the pH value of the reaction solution system of *C. vulgaris* and Cd was 10, the adsorption capacity of *C. vulgaris* was the greatest, and the adsorption efficiency of Cd was the highest. So we consider when this method is applied to the detection of heavy metals Cd in discharged industrial wastewaters, 0.05 mol L⁻¹ Na₂B₄O₇-NaOH buffer solution with a pH of 10 will be used to adjust the pH value of reaction solution to 10. This way the final pH of the reaction system can't be affected by the sample acidity.

3. Comment: Please present the study of the effect of matrix.

Response: Thank you very much for your comment. In this paper, we used *C. vulgaris* in the culture solution after 8 days of cultivation to enrich heavy metal Cd. Therefore, according to your suggestion, we have carefully considered that the matrix you said should refer to the matrix in the algae culture solution at this time. We think that the matrix in the culture solution at this time mainly includes some remaining nutrients in the culture medium, some substances secreted by the algae cells during the culture process, and some dead algae cells. According to the research results in the section "**3.2.1 Effects of solution environment of algae cells and contact time**" in our revised manuscript, after removing the original culture solution by centrifugation and washing, the adsorption capability of *C. vulgaris* re-dispersed in ultrapure water

was significantly lower than that of *C. vulgaris* in culture solution. Since *C. vulgaris* cells were centrifuged and washed in order to remove the original culture solution, the matrix in the culture solution was also removed, so this experimental result shows that the matrix in the original culture solution would not reduce the adsorption capability of *C. vulgaris* for Cd, on the contrary, it may increase the adsorption capability of *C. vulgaris* for Cd. Of course, it also may be because the centrifugation and washing processes affected the activity of *C. vulgaris*, and then the adsorption capability of *C. vulgaris* for Cd was affected. On the other hand, for the preconcentration experiment, the preconcentration process of heavy metal Cd was as follows: Firstly, a reaction solution was formed by mixing 25 mL of algae culture solution, 0.5 mL buffer solution and a certain amount of Cd test samples, and using ultrapure water to dilute to 40 mL, and then the subsequent adsorption reaction and preconcentration process were carried out. For this adsorption reaction process, the matrix in the *C. vulgaris* culture solution was present in the reaction solution. And the experimental results of "**3.2.3 Effect of initial Cd concentration**" in our revised manuscript also showed that under the optimal adsorption conditions, when the Cd concentration was in the range of 0.703-74.957 $\mu\text{g mL}^{-1}$, the adsorption efficiencies were more than 90%, and the highest adsorption efficiency could reach 99.31%. Such a high adsorption efficiency also indicates that the matrix in the culture solution would not affect the adsorption of Cd by *C. vulgaris*. So in our manuscript, we didn't further describe the effect of matrix.

4. Comment: The *C. vulgaris* could adsorb other inorganic species from water. In this way, a study of potential interferents need to be presented.

Response: Thank you very much for your comment. We are very sorry that in our original manuscript we didn't provide the experimental results of potential interferences of other inorganic species from water for the adsorption of Cd by *C. vulgaris*. Therefore, according to your comment, in our revised manuscript, we have carried out this experimental research, and the description of the experimental process

has been added to the second paragraph in section "**2.2 Preconcentration procedure**", the experimental results also have been supplemented as a section "**3.3 Influence of coexisting ions**" in our revised manuscript. These results indicated that even at high concentrations, those coexisting inorganic ions (Na^+ , K^+ , NH_4^+ , Ca^{2+} , Mg^{2+} , Fe^{3+} , Mn^{2+} , Zn^{2+} , NO_3^- , SO_4^{2-} and Cl^-) had no significant influence on the adsorption of Cd by *C. vulgaris*.

5. Comment: Inform how the detection limit was calculated.

Response: Thank you very much for your suggestion. According to IUPAC criteria, the calculation formula of detection limit is: $\text{LOD}=3\sigma/k$, where LOD is the detection limit, σ is the standard deviation of multiple measurements for blank sample, and k is the slope of the calibration curve. In this study, when the initial concentration of Cd was in the range of 0.703-74.957 $\mu\text{g mL}^{-1}$, for the obtained Cd-enriched thin samples, the net integral fluorescence intensity of Cd $\text{K}\alpha$ characteristic peak had a very good linear relationship with the initial concentration of Cd, and the linear calibration plot was $F=447.53061C+1193.35273$ with the linear correlation coefficient R^2 of 0.9979, where F was the net integral fluorescence intensity of Cd $\text{K}\alpha$ characteristic peak, and C was the initial concentration of Cd (the unit was $\mu\text{g mL}^{-1}$). So the slope of the calibration curve was 447.503061 $\text{mL } \mu\text{g}^{-1}$. For the blank filter membrane sample loaded *C. vulgaris* cells, the EDXRF spectrum measurement was repeated 11 times under the same experimental conditions, and the standard deviation (σ) of the integrated fluorescence intensity obtained from 11 repeated measurements was 9.7617. So according to the calculation formula of detection limit, the detection limit of this proposed method was $\text{LOD}=(3 \times 9.7617)/(447.503061 \text{ mL } \mu\text{g}^{-1})=0.0654 \mu\text{g mL}^{-1}$. In our revised manuscript, we have supplemented the sentences "***For the blank filter membrane sample loaded C. vulgaris cells, the EDXRF spectrum measurement was repeated 11 times under the same experimental conditions, and the standard deviation (σ) of the integrated fluorescence intensity obtained from 11 repeated measurements was 9.7617. And the slope (k) of the above linear calibration plot was***

447.503061 mL μg^{-1} . So according to the $3\sigma/k$ IUPAC criteria (σ is the standard deviation of multiple measurements for blank sample, k is the slope of linear calibration plot)," in line 23-32 in the section "3.5 Detection of Cd by EDXRF" in order to explain in detail how the detection limit was calculated according to the above calculation process.

6. Comment: The method was unable to determine cadmium in the water samples without adding the analyte. However, the detection limit of the method can be improved and decreased by varying, for example, the sample volume. Additional experiments need to be present with this intention.

Response: Thank you very much for your comment. In this paper, for the preconcentration method, the Cd thin samples were obtained by filtering 15 mL of the reaction solution, and the diameter of the Cd-enriched region on these formed Cd thin samples was 15.1 mm. Combined with this preconcentration method, the detection limit of this proposed EDXRF detection method for Cd was $0.0654 \mu\text{g mL}^{-1}$. Although this detection limit was lower than the maximum allowable discharge concentration of Cd in various industries wastewaters, it was higher than the standard limit for heavy metals Cd in other waters such as surface water, drinking water, and urban water supply, and so on. So as you mentioned, at present, the proposed method was unable to determine Cd in these water samples without adding the analyte Cd. However, for the preconcentration process, if we increase the volume of the reaction solution during filtration, and try to make the diameter of the Cd-enriched region on the Cd thin samples smaller, in this way, for the same water sample to be tested, we can obtain a Cd thin sample with a higher Cd mass per unit area by this improved preconcentration method. And this new Cd thin sample can obtain a stronger spectral signal by EDXRF spectrum measurement compared with the Cd thin sample obtained by the preconcentration method already used in this paper. Moreover, the slope (k) value in the linear calibration plot will be increased compared with the slope value in our manuscript. Therefore, the calculated detection limit will be lower. This is a very

interesting and worthwhile research work. At present, we have already preliminarily tried this research. But more in-depth and detailed research work will be carried out in the future, hoping to obtain a lower detection limit in order to meet the detection requirements of heavy metal Cd in some water samples such as surface water, drinking water, urban water supply, and so on.

7. Comment: The method is not sufficiently validated. For example: the results obtained need to be compared with a comparative method or a water certified reference material needs to be analyzed.

Response: Thank you very much for your suggestions. It is really true as Reviewer suggested that in our original manuscript, the proposed method was not sufficiently validated. So according to your suggestion, in our revised manuscript, we have used two certified Cd reference water samples, GBW(E)082822a-1 and GBW(E)082822b-2 to validate the accuracy of our proposed method, and the related descriptions and results have been supplemented in line 36-46 in section "**3.5 Detection of Cd by EDXRF**".

We have try our best to improve the manuscript and carefully revised our manuscript according to your comments and suggestions. So some changes have been made in the manuscript but they will not influence the content and framework of the paper. Once again, special thanks to you for your good comments and suggestions.

A list of the changes we have made to the original manuscript:

1. In abstract, the sentence "*C. vulgaris could directly and rapidly adsorb Cd(II) without adding any other reagents,*" in line 5 in our original manuscript has been corrected as "**C. vulgaris could directly and rapidly adsorb Cd²⁺ without any pre-treatment,**" in our revised manuscript. (see Abstract)
2. In abstract, the description "*the rapid detection and early warning for exceeding standard of Cd*" in line 22 in our original manuscript has been corrected as "**the rapid detection and early warning of excessive Cd**" in our revised manuscript. (see Abstract)
3. All the "*Cd(II)*" in our original manuscript have been corrected as "**Cd²⁺**" in our revised manuscript.
4. In section "**1. Introduction**", the sentence "*The entire detecting process is sophisticated.....*" in line 6 in the second paragraph in our original manuscript has been corrected as "**The entire detecting processes are usually sophisticated.....**" in our revised manuscript. (see Introduction)
5. In section "**1. Introduction**", the sentence "*In recent years, some preconcentration methods of heavy metals in water such as solid phase extraction has been used prior to.....*" in line 1-2 in the fourth paragraph in our original manuscript has been corrected as "**In recent years, some preconcentration methods of heavy metals in water such as solid phase extraction have been used prior to.....**" in our revised manuscript. (see Introduction)
6. In section "**1. Introduction**", the description "*..... some corresponding research works in this area*" in line 8 in the fifth paragraph in our original manuscript has been corrected as "**..... some related research works in this area**" in our revised manuscript. (see Introduction)
7. In the last paragraph of section "**1. Introduction**", the sentence "**Then the**

influences of different coexisting ions on the adsorption efficiency of Cd by *C. vulgaris* were investigated." was added in line 9 in our revised manuscript. (see **Introduction**)

8. In the last paragraph of section "**1. Introduction**", the description "..... *on the mixed cellulose membranes*" in our original manuscript has been corrected as "..... **on the mixed cellulose lipid microfiltration membranes**" in line 13 in our revised manuscript. (see **Introduction**)

9. In the last paragraph of section "**1. Introduction**", the description "..... *stability of XRF spectrometry*....." in our original manuscript has been corrected as "..... **stability of EDXRF spectrometry**....." in line 16 in our revised manuscript. (see **Introduction**)

10. In section "**2. Experimental**", the subheading "**2.1 Algal culture**" in our original manuscript has been corrected as "**2.1 Algae culture and characterization**" in our revised manuscript. (see **Experimental**)

11. At the end of section "**2.1 Algae culture and characterization**", the sentences "**All flasks were shaken 3 times and the state of *C. vulgaris* culture solution was observed daily.....the biomass of *C. vulgaris* in culture solution was calculated. Then *C. vulgaris* in the culture solution were used for adsorption and preconcentration of Cd in water samples**" have been added in line 14-30 in our revised manuscript in order to describe the experiments for characterizing *C. vulgaris* culture solution and determination of mass of adsorbent in the suspension. (see **Experimental**)

12. In section "**2.2 Preconcentration procedure**", the sentences "**Sodium borate ($\text{Na}_2\text{B}_4\text{O}_7 \cdot 10\text{H}_2\text{O}$) and sodium hydroxide (NaOH) were of analytical grade and purchased from Tianjin Recovery Fine Chemical Industry Research Institute (Tianjin, China). 0.1 mol L^{-1} NaOH solution, 2% (v/v) dilute HNO_3 solution, and 0.05 mol L^{-1} $\text{Na}_2\text{B}_4\text{O}_7$ -NaOH buffer solution with a pH of 10 were respectively prepared to adjust the pH values of reaction solutions.**" have been added in line

8-12 in our revised manuscript. (see **Experimental**)

13. In section "2.2 Preconcentration procedure", the sentence "*For the preconcentration of Cd, firstly, 25 mL of C. vulgaris culture solution and different volumes of Cd(II) stock solution were sequentially added into a 100 mL glass flask.*" in our original manuscript has been corrected as "**For the preconcentration of Cd, firstly, 25 mL of C. vulgaris culture solution, 0.5 mL of 0.05 mol L⁻¹ Na₂B₄O₇-NaOH buffer solution (pH 10) and different amounts (3.5, 7, 18, 56, 108, 230, 480, 595, 812 μL and 1.5, 3.4, 6, 11 mL) of Cd²⁺ stock solution were added into a 100 mL glass flask in sequence, then a certain volume of ultrapure water was added to dilute the reacton solution to 40 mL.**" in line 14-18 in our revised manuscript, and the sentence "*Under the same conditions, a series of blank samples were prepared in the same way by mixing 25 mL ultrapure water and the same volume of Cd(II) stock solution as above in 100 mL glass flasks. And Cd(II) concentration in the blank samples were measured by ICP-MS.*" in our original manuscript has been corrected as "**Under the same conditions, a series of 40 mL reference samples were prepared in the same way by mixing a certain volume of ultrapure water, 0.5 mL of 0.05 mol L⁻¹ Na₂B₄O₇-NaOH buffer solution (pH=10) and the same volume of Cd²⁺ stock solution as above (3.5, 7, 18, 56, 108, 230, 480, 595, 812 μL and 1.5, 3.4, 6, 11 mL) in 100 mL glass flasks. And Cd²⁺ concentrations in the reference samples were determined by ICP-MS as the initial concentrations of Cd²⁺ in the reaction solutions.**" in line 27-32 in our revised manuscript in order to clearly state the specific sample volume. (see **Experimental**)

14. In section "2.2 Preconcentration procedure", the sentences "**For the interference experiment of different coexisting ions on the adsorption the adsorption efficiency of Cd by C. vulgaris in the presence of different ions was calculated**" have been added in the last paragraph in our revised manuscript in order to describe the experimental process of potential interferences of other inorganic species from water for the adsorption of Cd by *C. vulgaris*. (see **Experimental**)

15. In section "2.4 Analysis of real water samples", the sentence "*The water samples*

were first filtered through *0.45 μm mixed cellulose membranes*" in our original manuscript has been corrected as "The water samples were first filtered through **mixed cellulose lipid microfiltration membranes with a pore size of 0.45 μm and a diameter of 50 mm (Shanghai Xingya Purification Material Factory, China)**....." in line 4-6 in our revised manuscript in order to provide more information about the mixed cellulose membranes. (see **Experimental**)

16. In section "2.4 Analysis of real water samples", the sentences "*The same preconcentration method was adopted to extract Cd(II) from the spiked real water samples. And the obtained Cd-enriched thin samples were measured by the EDXRF spectrometer.*" in our original manuscript has been corrected as "The same preconcentration method **as above** was adopted to extract **Cd²⁺** from the spiked real water samples **to form Cd-enriched thin samples. At the same time, Cd calibration standard samples and blank samples were prepared according to the above preconcentration procedure, and the concentrations of standard samples were determined by ICP-MS. And the obtained Cd-enriched thin samples corresponding real water samples and Cd calibration standard samples were measured by the EDXRF spectrometer under the same experimental conditions.**" in line 8-15 in our revised manuscript in order to supplement the detailed description of preparation of the calibration standards and blank. (see **Experimental**)

17. In section "3. Results and discussion", the subheadings "**3.1 Characterization of *C. vulgaris* culture solution**" and "**3.3 Influence of coexisting ions**" have been added in our revised manuscript, so the subheading numbers in the original manuscript have been adjusted in our revised manuscript. (see **Results and discussion**)

18. In section "**3.1 Characterization of *C. vulgaris* culture solution**" in our revised manuscript, the sentences "**During the cultivation process,..... And the biomass of *C. vulgaris* in culture solution was 0.8253 g L⁻¹**" have been added in order to supplement the characterization results of *C. vulgaris* culture solution and the determination results of mass of adsorbent in the suspension. (see **section 3.1**)

19. In section "3.2.1 Effects of solution environment of algae cells and contact time" in our revised manuscript, the sentences "**This may be that the *C. vulgaris* in the original culture solution had good activity state, so the adsorption capacity of *C. vulgaris* re-dispersed in ultrapure water for Cd was reduced.**" have been added in line 32-41 in order to supplement the discussion on why the use of the *C. vulgaris* in culture solution was more efficient. (see section 3.2.1)

20. In section "3.2.2 Effect of reaction solution pH " in our revised manuscript, the sentences "**For the mixture of *C. vulgaris* culture solution and Cd²⁺, the pH value of the reaction system was adjusted from 10 to 2.56 by dropwise adding 0.1 mol L⁻¹ NaOH or 2% (v/v) dilute HNO₃ solutions**" in line 3-6, "**And when the pH value of the reaction solution system of *C. vulgaris* culture solution and Cd was 10**" in line 21-22, and "**Therefore, the optimal pH of the reaction solution was 10. And in the subsequent study, for the process of adsorption and preconcentration of Cd from water samples, the reaction solution was adjusted to 10 by adding 0.5 mL of 0.05 mol L⁻¹ Na₂B₄O₇-NaOH buffer solution with a pH of 10**" in line 24-27 have been added in order to supplement the description about how to control the pH of the reaction solution. (see section 3.2.2)

21. In section "3.2.3 Effect of initial Cd concentration", the sentence "*This may be attributed to when the initial concentration*" in line 16 in our original manuscript has been corrected as "**This may be because when the initial concentration**" in our revised manuscript, and the sentence "*so the adsorption efficiency of Cd(II) decreased*" in line 21 in our original manuscript has been corrected as "**so the calculated adsorption efficiency of Cd²⁺ decreased**" in our revised manuscript. (see section 3.2.3)

22. In section "3.3 Influence of coexisting ions" in our revised manuscript, the sentences "**Due to the existence of a variety of inorganic ions in real environmental water samples These results indicated that even at high concentrations, those coexisting inorganic ions had no significant influence on the adsorption of Cd by *C. vulgaris*.**" and **Table 1** have been supplemented in order

to describe the experimental results of potential interferences of other inorganic species from water for the adsorption of Cd by *C. vulgaris*. (see section 3.3)

23. In section "3.5 Detection of Cd by EDXRF", the description ".....*integral fluorescence intensity of the blank mixed cellulose membrane.*" in line 11 in our original manuscript has been corrected as ".....**integral fluorescence intensity of the blank filter membrane sample loaded *C. vulgaris* cells.**" in our revised manuscript. (see section 3.5)

24. In section "3.5 Detection of Cd by EDXRF", the sentences "**For the blank filter membrane sample loaded *C. vulgaris* cells, the EDXRF spectrum measurement was repeated 11 times under the same experimental conditions, and the standard deviation (σ) of the integrated fluorescence intensity obtained from 11 repeated measurements was 9.7617. And the slope (k) of the above linear calibration plot was 447.503061 mL μg^{-1} . So according to the $3\sigma/k$ IUPAC criteria (σ is the standard deviation of multiple measurements for blank sample, k is the slope of linear calibration plot)**" have been added in line 23-29 in our revised manuscript in order to explain in detail how the detection limit was calculated. (see section 3.5)

25. In section "3.5 Detection of Cd by EDXRF", the sentences "**In addition, the accuracy of the proposed method for detection of Cd was evaluated by analyzing two certified reference water samples, GBW(E)082822a-1 (certified Cd concentration: 100 $\mu\text{g mL}^{-1}$) and GBW(E)082822b-2 (certified Cd concentration: 500 $\mu\text{g mL}^{-1}$). Prior to preconcentration, the reaction solution composed of 25 mL of *C. vulgaris* culture solution, 0.5 mL of $\text{Na}_2\text{B}_4\text{O}_7\text{-NaOH}$ buffer solution (pH 10) and 1 mL Cd certified reference water sample was diluted to 40 mL with ultrapure water. Then preconcentration and detection of Cd were performed according to the proposed method. And compared with the certified Cd concentrations of the two reference water samples, the relative errors of the detection results were 4.7% and 3.1%, respectively.**" have been added in line 36-46 in our revised manuscript in order to validate the accuracy of our proposed method. (see section 3.5)

26. The numbers of "Table1" and "Table2" in our original manuscript have been corrected as "Table2" and "Table3" in our revised manuscript. (see **Results and discussion**)

27. In section "**4. Conclusions**", the sentence "*the maximum adsorption efficiency could be obtained when the contact time was 1min*" in line 2-3 in our original manuscript has been corrected as "**the maximum adsorption efficiency could be obtained when the contact time was 1min and the pH of reaction solution was 10**" in our revised manuscript. (see **Conclusions**)

28. The format of the reference numbers in the main text of our revised manuscript has been modified. For example: "*cardiovascular system and reproductive system of human body will be seriously affected,¹*" and "*In addition, Cd has been known as a human carcinogen by the International Agency for Research on Cancer.^{2,3}*" in our original manuscript have been corrected as "**cardiovascular system and reproductive system of human body will be seriously affected [1],**" and "**In addition, Cd has been known as a human carcinogen by the International Agency for Research on Cancer [2,3].**" in our revised manuscript.

29. The format of the references in our revised manuscript has been modified according to the reference format in the *Royal Society Open Science* template.

30. In our revised manuscript, the data deposited in Dryad Digital Repository have been added as reference [32] in the reference list and cited in the data accessibility section.

31. Some grammar or spelling mistakes have been carefully corrected in our revised manuscript.